

# Foxa2 attenuates steatosis and inhibits the NF-κB/IKK signaling pathway in nonalcoholic fatty liver disease

Li Yang[1,*], Qiang Ma[2,*], Jiayu Chen[2], Xiangcai Kong[2], Xiaohui Yu[2] and Wei Wang[2]

[1] Northwest Minzu University, Lanzhou, Gansu, China
[2] Department of Gastroenterology, 940th Hospital of Joint Support Force, Lanzhou, Gansu, China
[*] These authors contributed equally to this work.

## ABSTRACT

**Objective**. Forkhead box a2 (Foxa2) is proven to be an insulin-sensitive transcriptional regulator and affects hepatic steatosis. This study aims to investigate the mechanism by which Foxa2 affects nonalcoholic fatty liver disease (NAFLD).

**Methods**. Animal and cellular models of NAFLD were constructed using high-fat diet (HFD) feeding and oleic acid (OA) stimulation, respectively. NAFLD mice received tail vein injections of either an overexpressing negative control (oe-NC) or Foxa2 (oe-Foxa2) for four weeks. HepG2 cells were transfected with oe-NC and oe-Foxa2 for 48 h before OA stimulation. Histological changes and lipid accumulation were assessed using hematoxylin-eosin staining and oil red O staining, respectively. Expression of Foxa2, NF-κB/IKK pathway proteins, lipid synthesis proteins, and fatty acid β-oxidation protein in HFD mice and OA-induced HepG2 cells was detected using western blot.

**Results**. Foxa2 expression was downregulated in HFD mice and OA-induced HepG2 cells. Foxa2 overexpression attenuated lipid accumulation and liver injury, and reduced the levels of aspartate aminotransferase, alanine aminotransferase, total cholesterol, or triglyceride in HFD mice and OA-induced HepG2 cells. Moreover, Foxa2 overexpression decreased the expression of lipid synthesis proteins and increased fatty acid β-oxidation protein expression in the liver tissues. Furthermore, overexpression of Foxa2 downregulated the expression of p-NF-κB/NF-κB and p-IKK/IKK in OA-induced HepG2 cells. Additionally, lipopolysaccharide (NF-κB/IKK pathway activator) administration reversed the downregulation of lipid synthesis proteins and the upregulation of fatty acid β-oxidation protein.

**Conclusion**. Foxa2 expression is downregulated in NAFLD. Foxa2 ameliorated hepatic steatosis and inhibited the activation of the NF-κB/IKK signaling pathway.

## INTRODUCTION

Nonalcoholic fatty liver disease (NAFLD) is a complex systemic disease that is characterized by hepatic lipid accumulation, lipotoxicity, insulin resistance, gut dysbiosis, and inflammation (*Tilg et al., 2021*). The global prevalence of NAFLD is reportedly as high

Corresponding author
Wei Wang, onewaywei@163.com

as 25% (*Younossi et al., 2016*). Currently, insulin resistance is considered a primary driver of NAFLD, as it disrupts lipid metabolism and leads to excessive deposition of fatty acids and triglycerides in hepatocytes (*Heo et al., 2019*; *Tilg, 2010*). Peroxidation of lipid components then leads to liver inflammation, and hepatocellular damage from this inflammation plays a crucial role in the progression from simple fatty liver to steatohepatitis (*Martín-Fernández et al., 2022*). However, there are currently no effective treatments for NAFLD, underscoring the need for molecularly targeted therapeutic strategies.

Forkhead box (Fox) transcription factors, Foxa1, Foxa2, and Foxa3, are essential for early mammalian development and organogenesis (*Moya et al., 2012*). Foxa2 has been identified as an insulin-sensitive transcriptional regulator that modulates gene expression involved in glucose and lipid metabolism pathways and improves insulin resistance in peripheral tissues (*Le Lay & Kaestner, 2010*; *Puigserver & Rodgers, 2006*; *Wolfrum et al., 2004*). Recent studies have shown that mutations and abnormal expression of Foxa2 are related to hepatic steatosis. For example, Foxa2 is permanently inactive and located in the cytoplasm of hepatocytes in hyperinsulinemia/obese mice, which favors the development of hepatic steatosis and insulin resistance (*Wolfrum et al., 2004*). The activity of Foxa2 is modulated through phosphorylation at the threonine 156 (Thr156) residue, a process mediated by the insulin/PI3K/Akt signaling pathway. This phosphorylation at the Thr156 position diminishes Foxa2 activity and initiates its export from the nucleus (*Wolfrum et al., 2003*). Foxa2 deficiency induces endoplasmic reticulum stress and alters the expression of developmental genes in human iPSC-derived hepatocytes (*Aghadi, Elgendy & Abdelalim, 2022*). Furthermore, Liu et al. showed that activation of Foxa2 expression ameliorated hepatic steatosis (*Liu et al., 2022*). The above data suggest that Foxa2 has the potential to inhibit the occurrence and progression of NAFLD. However, the mechanism through which Foxa2 acts is not fully understood.

The NF-κB/IKK signaling pathway serves as a bridge between Foxa2 and NAFLD. In NAFLD, activation of this pathway is associated with inflammation, oxidative stress, and insulin resistance—all key factors in the development of the disease (*Heo et al., 2019*). NF-κB activation increase the levels of pro-inflammatory factors, which contribute to the progression of liver injury (*Farrell et al., 2012*; *Popko et al., 2010*). NF-κB activation stimulates the generation of reactive oxygen species, causing oxidative damage to liver cells (*Zhong et al., 2021*). The NF-κB/IKK pathway activation also impairs insulin signaling, leading to insulin resistance and further exacerbate the progression of NAFLD (*Fawzy et al., 2021*). Therefore, we hypothesized that Foxa2 may improve NAFLD *via* regulating the NF-κB/IKK signaling pathway.

In this research, animal and cellular models of NAFLD were constructed to verify whether Foxa2 attenuates NAFLD progression by modulating the NF-κB/IKK pathway. Our research may offer a new strategy for the treatment of NAFLD.

## MATERIALS & METHODS

### Animals

Twenty-four male C57BL/6 mice (four-week-old), obtained from GemPharmatech Co. Ltd. (Nanjing, China), were used to build the NAFLD mouse model. They were maintained

in a controlled room with the temperature at 22–25 °C, a humidity of 55% ± 5%, and 12 h light/dark cycle. One week later, mice were randomly divided into four groups ($n = 6$): control, high-fat diet (HFD), HFD + overexpression negative control (oe-NC), and HFD + oe-Foxa2. Control mice were fed a normal chow and the rest were fed HFD (consisting of 60% fat, 20% protein, and 20% carbohydrate) (*Qiao et al., 2018*). After eight weeks, the HFD-fed mice received tail-vein injections of $1 \times 10^7$ TU/mL of lentiviral-packed oe-Foxa2 (NM_001291065.1) or oe-NC (Vector ID: VB010000-9389rbj; VectorBuilder, Guangzhou, China) every two weeks, totaling two injections (*Wang et al., 2017*). Four weeks post-injection, the mice were anesthetized with sodium pentobarbital (50 mg/kg) for blood collection from the abdominal aorta. They were then sacrificed by cervical dislocation and liver tissue was collected for the following experiments. The animal study protocol was approved by Ethics Committee of the 940th Hospital of Joint Support Force (2020KYLL004).

## Body and liver weight
After the feeding experiment, the weight of individual mice was weighed. The liver of each animal was removed and weighed, and the ratio of liver weight to body weight was calculated.

## Histological examination
The histological changes in liver tissues were observed using a hematoxylin-eosin (HE) kit (#C0105S; Beyotime, Shanghai, China) according to manufacturer's instruction. Simply, liver tissue samples were fixed in 10% formaldehyde for 48 h before undergoing a graded ethanol dehydration series (50%–100%). The samples were then cleared in xylene and embedded in paraffin. Sections of 4 μm thickness were prepared using a microtome (Leica, Wetzlar, Germany) after pre-cooling the tissue blocks at −20 °C. These sections were affixed to slides and baked at 65 °C for adhesion. Deparaffinization and rehydration followed, utilizing xylene and a descending alcohol series (100%–75%), respectively. Staining was conducted using hematoxylin and eosin solutions for cellular morphology and inflammation assessment. The slides were then cleared in xylene and sealed with neutral balsam. Microscopic imaging was subsequently performed to evaluate the tissue samples. Images were captured using a microscope (Olympus, Tokyo, Japan).

## Cell culture and treatment
Human HepG2 cells were obtained from American Type Culture Collection (Manassas, VA, USA) and maintained in Dulbecco's Modified Eagle Medium containing 10% fetal bovine serum (Gibco, Waltham, MA, USA) under 5% $CO_2$ at 37 °C. HepG2 cells were transfected with oe-Foxa2 (NM_021784.5) or oe-NC (VectorBuilder) using HighGene transfection reagent (ABclonal, Wuhan, China) for 48 h. Then, the medium was changed and supplemented with oleic acid (OA; 0.5 mM) to induce the cellular NAFLD model. After 16 h, cells were harvested for subsequent analysis. After transfected with oe-Foxa2 and treated with OA, cells were treated with NF-κB/IKK pathway activator lipopolysaccharide (LPS, 1 μg/mL) for 12 h to validate the effect of the NF-κB/IKK pathway in hepatic steatosis.

## Oil red O staining

Oil red O staining was performed on an oil red O staining kit (#C0157S; Beyotime, Shanghai, China) according to the manufacturer's illustration. For animals, frozen liver tissues were cut into 15 μm slices. For HepG2 cells, harvested cells were fixed with 4% paraformaldehyde. Then, slices or cells were dipped in oil red O solution for 20 min, washed with phosphate-buffered saline (PBS), and took photos with a microscope (Olympus, Tokyo, Japan). Afterward, the oil red O reagent was dissolved by 100% isopropanol. The absorbance value was examined at 490 nm.

## Biochemical and ELISA analysis

The levels of aspartate aminotransferase (AST) and alanine aminotransferase (ALT) in serum were detected using a biochemical auto-analyzer (BIOBASE, Jinan, China).

Liver tissues were first homogenized in a protein extraction buffer containing protease inhibitors. HepG2 cells were washed with PBS and then lysed with RIPA lysis buffer supplemented with protease and phosphatase inhibitors. After homogenization or cell lysis, the samples were centrifuged at 12,000 g for 15 min at 4 °C. The supernatant was then collected for further analysis. The levels of total cholesterol (TC; #YFXEM00593; Yifei Xue Biotechnology, Nanjing, China) and triglyceride (TG; #ml037202; mIbio, Shanghai, China) in liver tissues or HepG2 cells were measured following ELISA kits instruction.

## Quantitative real-time PCR (qRT-PCR)

Total RNA was extracted from cells using the TRIzol reagent (Invitrogen, Waltham, MA, USA) and the RNA concentration was measured by the Nano-Drop 8000 (Thermo, Waltham, MA, USA). FastKing gDNA Dispelling RT reagent Kit (TIANGEN, Beijing, China) was used to synthesis cDNA from 1 μg of total RNA. qRT-PCR was performed on a CFX96 Touch instrument (Bio-Rad, Hercules, CA, USA) with SYBR Green PCR Master Mix (Roche Applied Sciences, Barcelona, Spain). Each well (20 μL PCR reaction volume) included 10 μL of SYBR Green PCR Master Mix, 1 μL of cDNA, 2 μL of primer pair mix (10 μM each primer) and 7 μL of DNAse/RNAse-free H2O. Each sample was prepared in three replicates. The reaction conditions were as follows: 95 °C for 3 min, 40 cycles of 95 °C for 12 s and 62 °C for 40 s. The relative expression of Foxa2 was calculated by a $2^{-\Delta\Delta Ct}$ method and normalized to GAPDH. The primer sequences are as follows: Foxa2 (NM_021784.5) forward 5′-TGCACTCGGCTTCCAGTATG-3′ and reverse 5′-CGT GTT CAT GCC GTT CAT CC-3′, PCR product length 143 bp, and GAPDH (NM_001357943.2) forward 5′-CAT GGG TGT GAA CCA TGA GAA-3′ and reverse 5′-GGC ATG GAC TGT GGT CAT GAG-3′, PCR product length 145 bp.

## Western blotting

Collected liver tissues and cells were lysed in RIPA buffer (Beyotime, Shanghai, China). After gel electrophoresis, samples were transferred onto membranes, blocked with 5% fat-free milk for 1 h, and incubated with primary antibodies at 4 °C for 12 h. After washing, membranes were incubated with secondary antibody (1:2000; ab205718; Abcam, Cambridge, UK) at 25 °C for 1 h. Protein bands were observed using an enhanced chemiluminescence kit (APPLYGEN, Beijing, China). The band intensities were quantified

using ImageJ software. The quantification was normalized to the intensity of the GAPDH band to account for potential loading variations. The primary antibodies including carnitine palmitoyl transferase 1α (CPT1α; 1:1000; DF12004), Foxa2 (1:1000; DF13363), NF-κB (1:2000; AF5006), p-NF-κB (1:2000; AF2006), IKK (1:2000; AF6014), and p-IKK (1:2000; AF3013) were purchased from Affinity Biosciences (JIangsu, China); acetyl-CoA carboxylase (ACC; 1:1000; 3676) and GAPDH (1:1000; 2118) were purchased from Cell Signaling Technology (Waltham, MA, USA); fatty acid synthase (FAS; 1:1000; E-AB-40063) was purchased from Elabscience Biotechnology Co., Ltd (Wuhan, China).

## Statistical analysis

Results data were exhibited as mean with standard deviations. Statistical analyses were carried out using GraphPad Prism 8.0 software. For comparisons involving only two groups, an unpaired $t$-test was performed. For comparisons among multiple groups, one-way analysis of variance followed by Tukey's test were conducted. $P < 0.05$ indicates results data significantly different.

## RESULTS

### Foxa2 expression is decreased in NAFLD mice

Foxa2, an insulin-sensitive transcriptional regulator, has a role in regulating hepatic lipid metabolism and ketogenesis (*Liu et al., 2022*; *Wolfrum et al., 2004*). In addition, a previous study showed that Foxa2 activation ameliorates hepatic steatosis (*Liu et al., 2022*). In this study, we built an HFD-induced NAFLD mouse model to examine whether Foxa2 overexpression could ameliorate NAFLD. HFD significantly increased body and liver weight and the ratio of liver/body weight as well as fasting blood glucose (FBG) of mice, whereas decreased Foxa2 expression ($P < 0.01$). However, these changes were effectively mitigated by overexpressing Foxa2 ($P < 0.05$; Figs. 1A–1C). These results suggest that Foxa2 may be a potential therapeutic target for ameliorating NAFLD symptoms.

### Foxa2 overexpression reduces hepatic steatosis in NAFLD mice

Hepatic steatosis is a key histological feature of NAFLD (*Pan et al., 2021*). The serum levels of AST and ALT in NAFLD mice were higher than those in control mice ($P < 0.01$). Similar trends of TG and TC were also observed in the liver tissues of NAFLD mice ($P < 0.01$). In contrast, oe-Foxa2 transfection suppressed the levels of AST, ALT, TG, and TC in HFD-stimulated NAFLD mice ($P < 0.01$; Figs. 2A–2B). HE staining revealed that the hepatocytes of NAFLD mice were filled with a large number of white vacuoles, hepatocyte ballooning, and inflammatory cell aggregation. oe-Foxa2 treatment resulted in a significantly smaller number of white vacuoles, occasional small vesicular fat vacuoles, and no inflammatory cell aggregation in mice. Oil Red O staining showed that the accumulation of lipid drops was reduced after oe-Foxa2 treatment (Fig. 2C), suggesting that Foxa2 could suppress hepatic steatosis in NAFLD mice.

### Foxa2 regulates hepatic lipid metabolism in NAFLD mice

To explore the mechanism by which Foxa2 regulates lipid metabolism in NAFLD mice, the expression of lipid synthesis proteins FAS and ACC and fatty acid β-oxidation protein

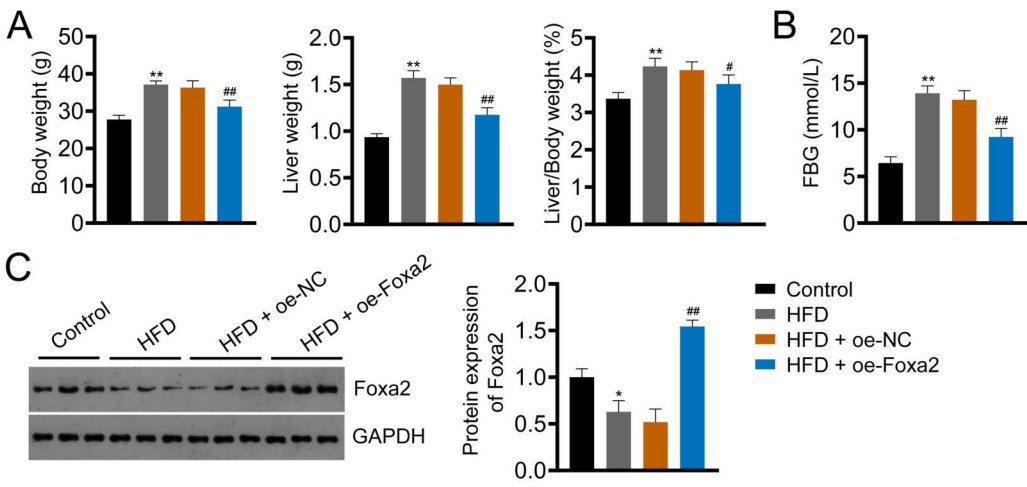

**Figure 1  Forkhead box a2 (Foxa2) expression is decreased in nonalcoholic fatty liver disease (NAFLD) mice.** (A) Body- and liver- weight and the ratio of body/liver weight. (B) Fasting blood glucose (FBG). (C) Protein expression of Foxa2 in liver tissues of each group of mice. C57BL/6 mice were fed with normal diet and high-fat diet (HFD) for eight weeks. Then, HFD mice were injected with overexpression negative control (oe-NC) and Foxa2 (oe-Foxa2) in the tail vein once every two weeks for four weeks. $*P < 0.05$ and $**P < 0.01$ compared with Control; $\#P < 0.05$ and $\#\#P < 0.01$ compared with HFD.

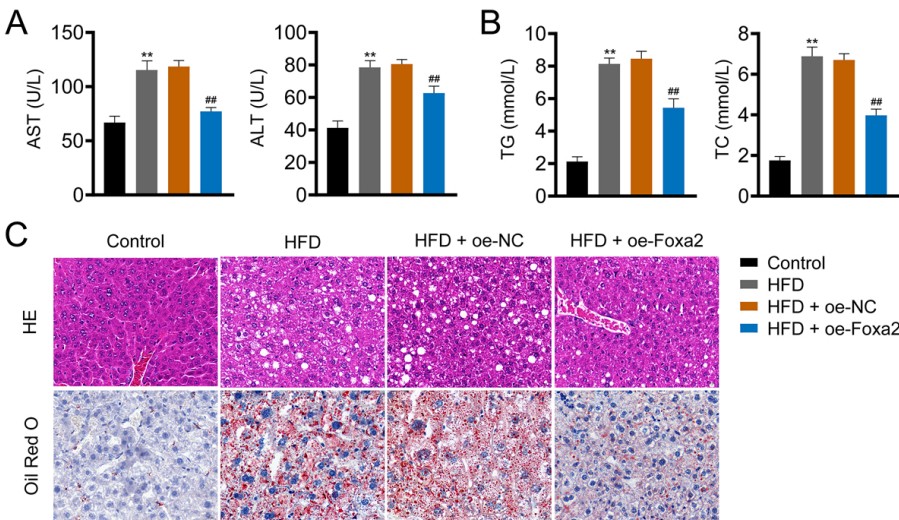

**Figure 2  Foxa2 overexpression reduces hepatic steatosis in NAFLD mice.** (A) The levels of aspartate aminotransferase (AST) and alanine aminotransferase (ALT) in serum. (B) The levels of total cholesterol (TC) and triglyceride (TG) in liver tissues. (C) Representative images of hematoxylin-eosin (HE) oil red O staining; scale bar: 20 μm. $**P < 0.01$ compared with Control; $\#\#P < 0.01$ compared with HFD.

CPT1α were measured by western blotting. The results showed that Foxa2 overexpression reduced HFD-stimulated FAS and ACC upregulation and CPT1α downregulation ($P < 0.05$; Fig. 3), indicating that Foxa2 inhibits lipid synthesis and facilitates fatty acid β-oxidation, which in turn reduces lipid deposition in the liver.
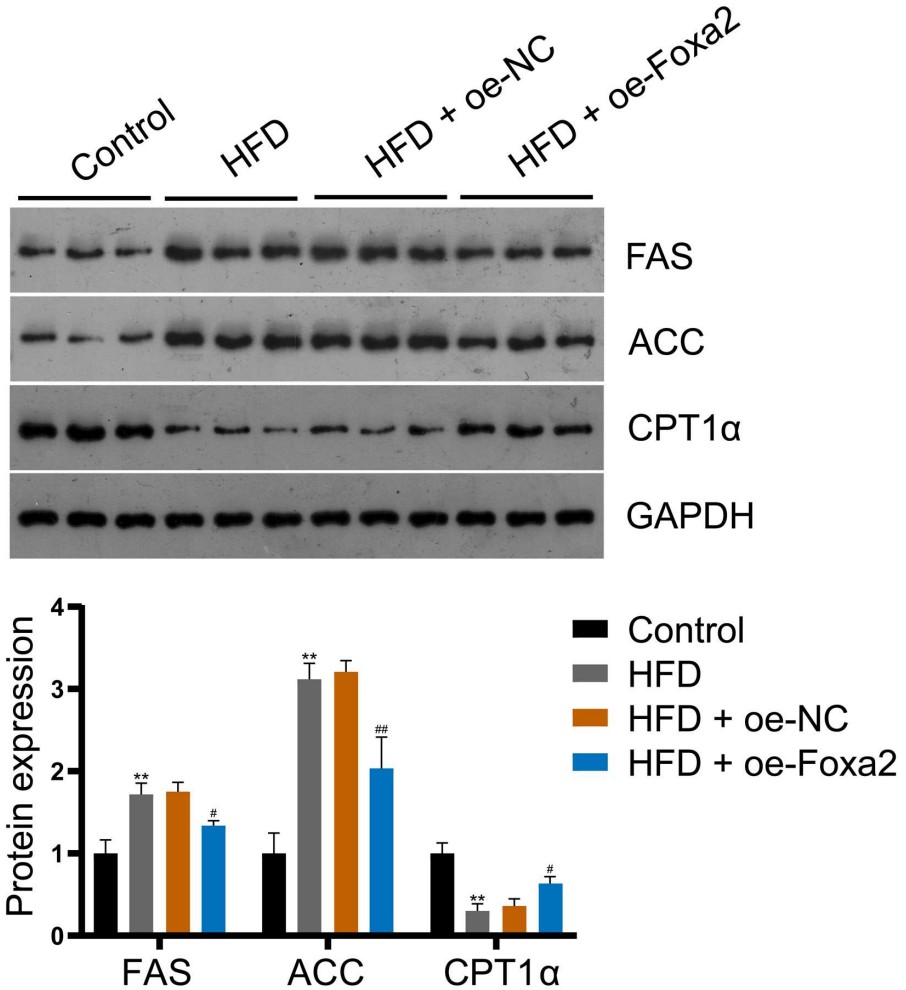

**Figure 3** **Foxa2 regulates hepatic lipid metabolism in NAFLD mice.** Protein expression of lipid synthesis proteins fatty acid synthase (FAS) and acetyl-CoA carboxylase (ACC) and fatty acid β-oxidation protein carnitine palmitoyl transferase 1α (CPT1α). **$P < 0.01$ compared with Control; #$P < 0.05$ and ##$P < 0.01$ compared with HFD.

## Foxa2 overexpression inhibits lipid accumulation in OA-induced HepG2 cells

To further investigate the effect of Foxa2 on NAFLD, we constructed an OA-induced NAFLD cellular model. Western blotting and qRT-PCR showed that OA significantly decreased Foxa2 protein and mRNA expression, which was reversed by Foxa2 overexpression ($P < 0.01$; Figs. 4A–4B). Oil red O staining exhibited that lipid accumulation was obvious in OA-induced HepG2 cells. Also, oe-Foxa2 transfection significantly reduced the elevated lipid content induced by OA treatment ($P < 0.01$; Fig. 4C). Next, the levels of TG and TC were enhanced in OA-induced HepG2 cells, which were suppressed by transfecting oe-Foxa2 ($P < 0.05$; Fig. 4D). Finally, the upregulation of FAS and ACC and the downregulation of CPT1α in OA-stimulated HepG2 cells were reversed after Foxa2

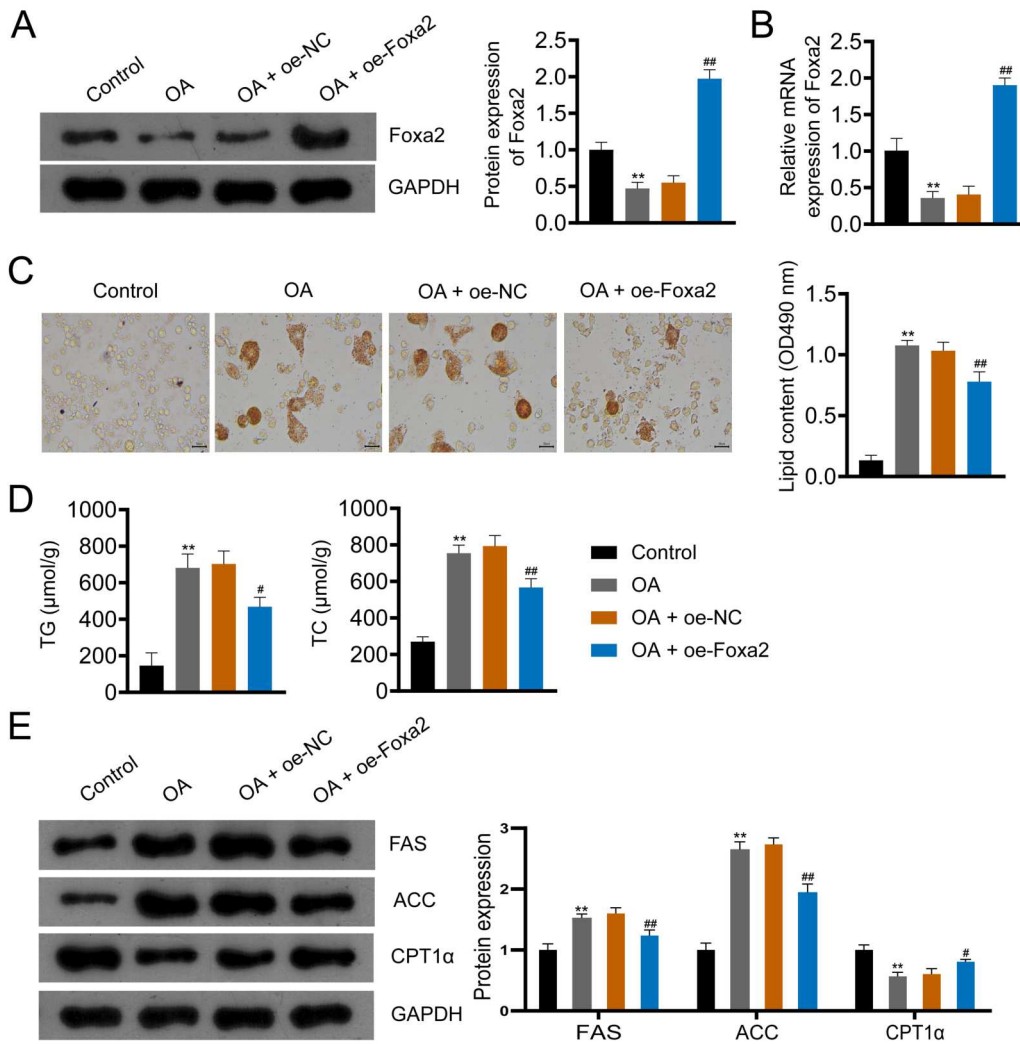

**Figure 4  Foxa2 overexpression inhibits lipid accumulation in OA-induced HepG2 cells.** (A–B) The protein and mRNA expression of Foxa2. (C) Oil red O staining; scale bar: 50 μm. (D) TG and TC levels were detected using ELISA kits. (E) Protein expression of FAS, ACC, and CPT1α were detected by western blotting. Before being treated with oleic acid (OA; 0.5 mM), HepG2 cells were transfected with oe-NC and oe-Foxa2. ** $P < 0.01$ compared with Control; # $P < 0.05$ and ## $P < 0.01$ compared with OA.

overexpression ($P < 0.05$; Fig. 4E). The *in vitro* results corroborate the *in vivo* findings and indicate that Foxa2 is a critical regulator of lipid metabolism in NAFLD.

## Foxa2 may ameliorate hepatic steatosis *via* inhibiting NF-κB/IKK pathway

NF-κB/IKK signaling pathway has been reported involved in steatosis (*Heida et al., 2021*; *Heo et al., 2019*). To verify that Foxa2 attenuates steatosis through the NF-κB/IKK pathway, OA-induced HepG2 cells were treated with oe-Foxa2 and LPS (NF-κB/IKK pathway activator). Western blotting suggested that OA significantly increased phosphorylation levels of NF-κB and IKK ($P < 0.01$). oe-Foxa2 transfection markedly downregulated the

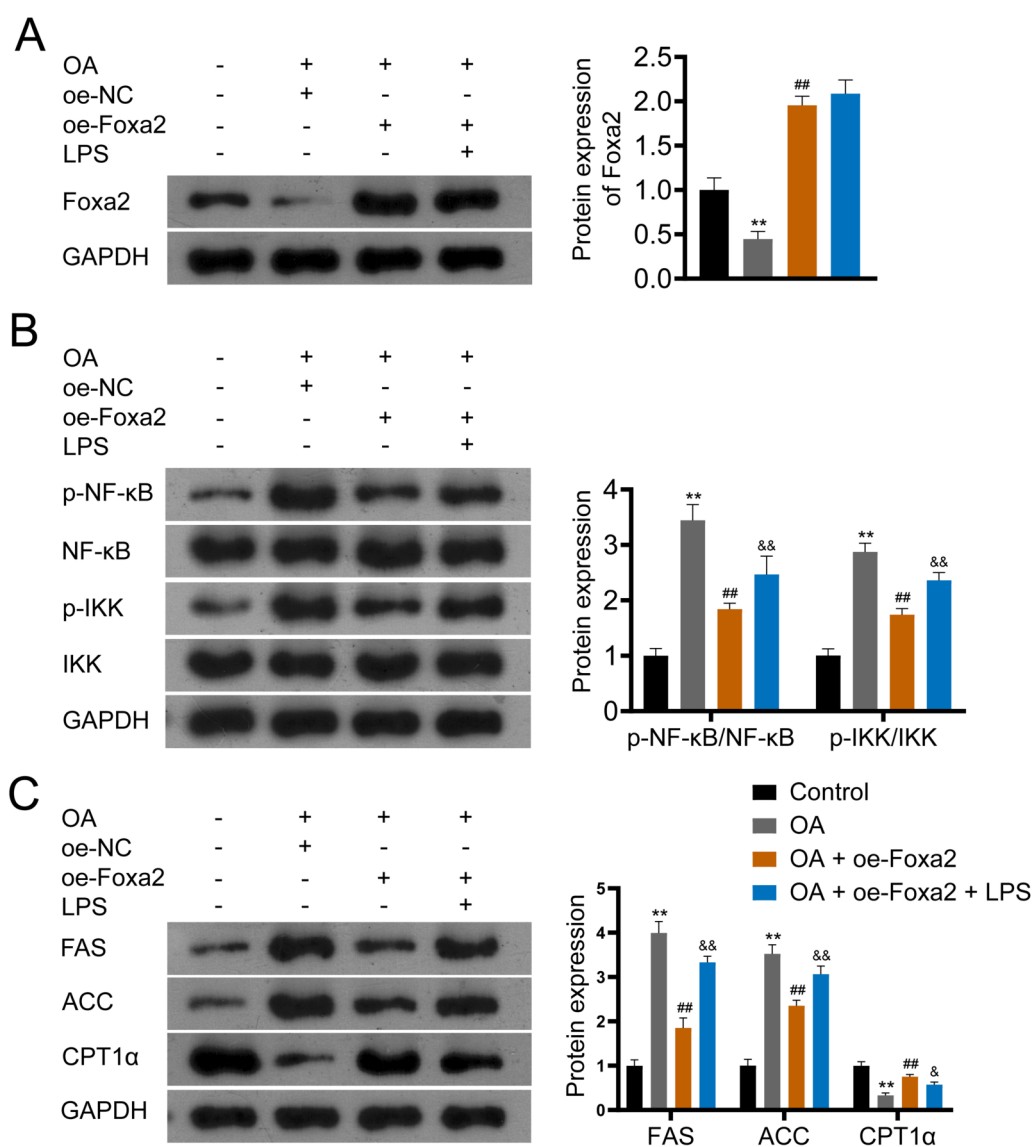

**Figure 5  Foxa2 ameliorates hepatic steatosis *via* inhibiting NF-κb/IKK pathway.** (A) Protein expression of Foxa2. (B) Protein expression of p-NF-κB, NF-κB, p-IKK, and IKK. (C) Protein expression of FAS, ACC, and CPT1α were detected by western blotting. HepG2 cells were treated with OA, oe-Foxa2, and lipopolysaccharide (LPS, 1 μg/mL). **$P < 0.01$ compared with Control; ##$P < 0.01$ compared with OA. &$P < 0.05$ and &&$P < 0.01$ compared with OA + oe-Foxa2 + LPS.

expression of p-NF-κB/NF-κB and p-IKK/IKK in OA-treated HepG2 cells, which was reversed by LPS administration. ($P < 0.01$; Figs. 5A–5B). Subsequently, western blotting also showed that oe-Foxa2-induced downregulation of FAS and ACC was reversed after LPS addition ($P < 0.01$). Conversely, LPS addition attenuated oe-Foxa2-induced CPT1α upregulation ($P < 0.05$; Fig. 5C). The above results suggest that Foxa2 may ameliorate hepatic steatosis by inhibiting the NF-κB/IKK pathway.

## DISCUSSION

In this study, we demonstrated that Foxa2 is important for the development of NAFLD. In both *in vivo* and *in vitro* models, Foxa2 overexpression attenuated lipid accumulation in hepatocytes and inhibited lipid synthesis in the liver, thereby promoting fatty acid β-oxidation. Furthermore, in an *in vitro* model, the attenuation of NAFLD progression by Foxa2 may be associated with the inhibition of the NF-κB/IKK pathway.

The "second hit" theory suggests that the pathogenesis of NAFLD is closely related to insulin resistance; insulin resistance is central to the development and progression of NAFLD (*Cotter & Rinella, 2020*; *Manne, Handa & Kowdley, 2018*; *Yang et al., 2022*). Foxa2 is an insulin-sensitive transcriptional regulator that plays an important role in liver development and metabolic functions (*Aghadi, Elgendy & Abdelalim, 2022*). *Wolfrum et al. (2004)* reported a study in mice with hyperinsulinemia caused by insulin resistance. In these mice, Foxa2 is retained in the cytoplasm in an inactive state. This inactivation disrupts the transcription of genes involved in lipid metabolism, resulting in lipid accumulation in the liver. Furthermore, the loss of specificity of the transcription factor Foxa2 in the liver leads to premature liver aging, increased hepatic lipogenesis and age-related obesity (*Bochkis, Shin & Kaestner, 2013*). In our study, we found that Foxa2 expression was downregulated in NAFLD and OA-induced HepG2 cells. All these indicate that Foxa2 may be a potential target for NAFLD treatment.

Liver enzymes, specifically ALT and AST, are commonly considered as markers of non-specific liver injury or hepatocellular inflammation (*Babu et al., 2021*). In patients with NAFLD, the concentrations of liver enzymes (AST and ALT) are elevated (*Giannini, Testa & Savarino, 2005*). In concordance with a previous study (*Liang et al., 2022*; *Zhang et al., 2023*), we found that AST and ALT levels were increased in NAFLD mice and OA-induced HepG2 cells, which were weakened by oe-Foxa2 transfection. A previous study reported that Foxa2 stimulates fatty acid oxidation, increases TG secretion, and reduces hepatic TG accumulation in obese animals (*Wan et al., 2011*). Here, Foxa2 overexpression significantly reduced the levels of TG and TC in NAFLD mice and OA-induced HepG2 cells. Furthermore, HE and oil red O staining showed that Foxa2 overexpression alleviated hepatocyte ballooning, inflammatory cell aggregation, and lipid droplet formation. Our results suggest that Foxa2 could alleviate lipid accumulation in NAFLD mice and OA-induced HepG2 cells.

Disorders of lipid metabolism contribute to the progression of NAFLD (*Xiao et al., 2023*). Accelerated lipid consumption and/or inhibited lipid synthesis are considered effective strategies for reducing hepatic lipid accumulation (*Fang et al., 2019*). ACC and FAS are key enzymes in fatty acid synthesis. He et al. found increased expression of FAS and ACC in OA-treated HepG2 cells (*He et al., 2017*), which was consistent with our findings. Fatty acid β-oxidation occurs mainly in the mitochondria and is regulated by PPARα and PGC1α (*Chen et al., 2019*). A large number of genes involved in fatty acid β-oxidation, such as CPT1α, are usually regulated by PPARα activation, which reduces hepatic lipid levels (*Pan et al., 2021*). In the current study, we found that CPT1α was decreased in NAFLD mice and OA-induced cells, which was the same as in a previous

study (*Pan et al., 2021*). Moreover, Wang and colleagues indicates that the activation of the AMPK/Foxa2 pathway can upregulate the expression of medium-chain acyl-CoA dehydrogenase, thereby promoting fatty acid oxidation and reducing lipid accumulation in nonalcoholic steatohepatitis (*Wang et al., 2022*). In our study, overexpression of Foxa2 reversed the upregulation of FAS and ACC and the downregulation of CPT1α in NAFLD mice and OA-stimulated HepG2 cells. These results suggest that Foxa2 improves hepatic steatosis to inhibit the progression of NAFLD.

In the resting state of cells, NF-κB is bound to its inhibitory protein (IκB) and exists in the cytoplasm in an inactive form (*Di Paola et al., 2018*; *Yamashita, Yoshida & Hayashi, 2016*). When cells are stressed or stimulated by microbes, viruses, and pro-inflammatory factors, IKK phosphorylation is activated, catalyzing IκB phosphorylation and consequently protein degradation (*Bai et al., 2017*). The free NF-κB becomes activated and enters the nucleus to regulate the transcription of immune-related receptors, cytokines, adhesion molecules, and other target genes, causing tissue damage (*Fattori et al., 2017*). It is well documented that NF-κB is essential in regulating inflammatory signaling pathways in the liver and inhibition of NF-κB reduces the extent of liver injury (*Mohammad et al., 2021*). In our study, we found that the levels of p-NF-κB/NF-κB and p-IKK/IKK were increased in OA-stimulated HepG2 cells, which were attenuated by Foxa2 overexpression. These suggest that the NF-κB/IKK pathway is activated in NAFLD and that Foxa2 overexpression inhibits the activation of the NF-κB/IKK pathway. Furthermore, oe-Foxa2-induced FAS and ACC downregulation and CPT1α upregulation was reversed by LPS (NF-κB/IKK pathway activator) treatment, which indicates that Foxa2 attenuates hepatic steatosis may relate to the inhibition of NF-κB/IKK pathway.

## CONCLUSIONS

In conclusion, we demonstrated that Foxa2 ameliorates hepatic steatosis in NAFLD and suppresses NF-κB/IKK signaling pathway activation in OA-stimulated HepG2 cells, thus presenting a new therapeutic target for NAFLD treatment. However, our study has some limitations, notably that our research is focused solely on the NF-κB/IKK pathway. It is important to note that further studies are required to explore the potential effects of Foxa2 through other mechanisms. This study both supplements and enriches the existing body of knowledge on Foxa2, thereby presenting a novel avenue for NAFLD treatment.

### Funding

This work was supported by the Gansu Digestive System Critical Diseases Clinical Medical Research Center Project (No. 20JR10RA017) and the Gansu Natural Science Foundation (No. 21JR11RA004). The funders had no role in study design, data collection and analysis, decision to publish, or preparation of the manuscript.

### Grant Disclosures

The following grant information was disclosed by the authors:

The Gansu Digestive System Critical Diseases Clinical Medical Research Center Project: 20JR10RA017.
The Gansu Natural Science Foundation: 21JR11RA004.

## Competing Interests

The authors declare there are no competing interests.

## Author Contributions

- Li Yang conceived and designed the experiments, analyzed the data, authored or reviewed drafts of the article, and approved the final draft.
- Qiang Ma conceived and designed the experiments, analyzed the data, authored or reviewed drafts of the article, and approved the final draft.
- Jiayu Chen performed the experiments, authored or reviewed drafts of the article, funding acquisition, and approved the final draft.
- Xiangcai Kong performed the experiments, prepared figures and/or tables, and approved the final draft.
- Xiaohui Yu analyzed the data, prepared figures and/or tables, authored or reviewed drafts of the article, and approved the final draft.
- Wei Wang conceived and designed the experiments, analyzed the data, authored or reviewed drafts of the article, funding acquisition, and approved the final draft.

## Animal Ethics

The following information was supplied relating to ethical approvals (i.e., approving body and any reference numbers):

940th Hospital of Joint Support Force (2020KYLL004)

## Data Availability

The raw data is available in the Supplemental Files.

## Supplemental Information

Supplemental information for this article can be found online at http://dx.doi.org/10.7717/peerj.16466#supplemental-information.

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
