# Peer review of "Foxa2 attenuates steatosis and inhibits the NF-κB/IKK signaling pathway in nonalcoholic fatty liver disease"

_PeerJ, doi:10.7717/peerj.16466_

## Round 0.1 · original submission · Major Revisions

Please carefully read the comments and address the quesitons from the reviewers.

Reviewer 1 ·

Basic reporting

This manuscript described a transcription factors FOX2 could ameliorates hepatic steatosis via inhibiting the NF-kappa B/IKK signaling pathway. However, I think there might be some logical connection problems .
1 Line 49 "new strategy" is a large conception. To better leading to the following paragraph, please use add precise qualifier for the type of strategy.
2 Line122 There was a garbled character before "FastKing"
3 The format was slightly confused. Two ends of the alignment should be noticed.
4 Please check all the tense of this manuscrpit.

Experimental design

Line 88 Please use words like sacrifice instande with execute for respecting animal welfare.

Validity of the findings

The results demonstrated the authors' hypothesis. However, the writing of result part was a mess. It read like a lab report without any detail instruction of each figures. Please rewrite this part.

Additional comments

no comment

·

Basic reporting

This paper reports on the effects of Foxa2 overexpression on alleviating lipid accumulation in mouse liver in vivo and in the HepG2 hepatoma cell line in vitro. It concludes that the effect of Foxa2 on steatosis is mediated via inhibition of NFKB signalling. The latter claim is not supported / overstated.

A limitation for a study focusing on Foxa2 is that there is no clear statement that Foxa2 is inactivated by Akt phosphorylation and exclusion from the nucleus.

A clear account of whether the negative control is an empty vector lacking Foxa2 or siRNA/shRNA is kacking.

Some statements in the Introduction and elsewhere are misleading, for example the first sentence that NAFLD is not associated with liver damage does not concur with later statements. Likewise, the 2nd sentence of the Discussion is not correct because there was no measure of de novo lipogenesis. Likewise the Aghadi reference did not measure hepatic steatosis because ir was a cellular study.

Experimental design

Details of the NC and overexpression vectors are essential to clarify whether the former was shRNA and whether the latter was human or mouse (gene reference lacking) or constiutively active.

Was the Immunoblotting analysis on whole lysate, supetnatant or pellet fraction?

Was any attempt made to measure nuclear (active) and cytoplasmic (inactive) fractions?

Validity of the findings

For the in vivo studies data was from n=6 for blood glucose and liver lipids and n=3 for western blot. The data file gives n=1 for WB, can the additional 2 sets be included?

Reviewer 3 ·

Basic reporting

The manuscript entitled “ Foxa2 attenuates steatosis in nonalcoholic fatty liver disease via inhibiting the NF-ĸB/IKK signaling pathway”, I found the manuscript is not adjusted in word format, No good presentation or justified text, the manuscript many punctuation errors and typos. The manuscript needs a substantial language editing for grammatical and scientific presentation.

Experimental design

Methods
- Add the reference of HFD preparation
- Write on the rationale for selecting the dose of oe-Foxa2.
- How did you euthanized mice?
- The method of Histological examination should be explained in detail.
- How did the primer sequences of genes were generated, accession number and the size ?

Validity of the findings

Abstract
- “ proved an insulin-sensitive” revise, “ proved to be an insulin-sensitive”
- “nonalcoholic” or “ non-alcoholic” Should be the same allover the manuscript.
- The method presented in the abstract is not clear and feel not match with that in the methods due to bad presentation.
- Line 26 : How did you measured the histopathological lesions?

Additional comments

Discussion
- Line 213 : Cite this research which is the most recent to record these findings “ file:///E:/my%20papers%202018/Gene%202023.htm”
- Line 215 : “A study has shown that in hyperinsulinemia mice caused by insulin resistance, Foxa2 is localized in the cytoplasm in an inactive state, and transcription of genes related to lipid metabolism is perturbed, leading to lipid accumulation in the liver” , I really can’t catch what you actually mean, you should fragment the sentence or rewrite.
- The authors need to address in the discussion why the “AST and ALT levels” were increased in NAFL group and compare these results with other published papers in this regard.
- Also , why Foxa2 overexpression alleviated hepatocyte ballooning, inflammatory cell aggregation, and lipid droplet formation?????? , compare your obtained results.
- The Conclusion of the study is very short and not well presented, You should add the limitations of the study.

Reviewer 4 ·

Basic reporting

The manuscript aims to elucidate the role of Foxa2 in nonalcoholic fatty liver disease (NAFLD) and explores its potential as a therapeutic target. The study is well-designed and the objectives are clear.

Experimental design

no comment

Validity of the findings

no comment

Additional comments

Major Comments:
Introduction:
a. While a comprehensive background is provided on NAFLD and Foxa2, the transition from one topic to another can be more smooth.
b. The introduction of NF-κB/IKK in the later part feels slightly abrupt. Consider reorganizing to create a more seamless flow between topics.

Methods:
a. Gender of the mice used should be specified to provide a clearer context.
b. It would be helpful to mention the duration and dosage of the high-fat diet feeding.
c. The methodology of Foxa2 overexpression for animal experiments should be elaborated. How were the overexpression constructs prepared, and what was the efficiency of transfection/transduction?
d. Clarify how the sections were analyzed post-staining in histological examination.
e. Specify the passage number or range for HepG2 cells to ensure reproducibility, as cell behavior might vary with passage number.
f. Specify how samples were prepared from liver tissues or HepG2 cells for ELISA analysis.
g. Explain how the band intensities were quantified, and mention the software or method used in western blotting.
h. The authors should include relevant statistical analyses to show differences between two groups and among multiple groups.

Results:
Were there any observable side effects from Foxa2 overexpression in the models used?

Discussion:
The potential clinical implications of these findings should be discussed in more depth. How does this study contribute to the existing body of knowledge on NAFLD and its management?

Minor Comments:
a. The abstract could be restructured for clarity, particularly the methods and results sections, ensuring they succinctly convey the critical points of the study.
b. The manuscript will benefit from a careful language review to correct any grammatical errors and enhance clarity. For example, Line 48 could be rephrased as: "There is currently no effective treatment for NAFLD, underscoring the need for new therapeutic strategies."
c. Check for consistent use of italics or specific styles for gene and protein names in line with journal guidelines.

---

## Round 0.2 · Major Revisions

Please carefully read the comments from Reviewer 2 and provide your responses accordingly.

Reviewer 1 ·

Basic reporting

no comment

Experimental design

no comment

Validity of the findings

no comment

Additional comments

no comment

·

Basic reporting

Three main points need yo be attended to:

1. Insufficient background context- The introduction should refer to the role of Akt phosphorylation of Foxa-2 because this is the basis of insulin-senitive regulation of transcription by Foxa-2.

2. The conclusion and title that Foxa2 regulates steatosis “via” NFKb is not supported by the data because the NFKB analysis was not done in mice but in the hepatoma cell line. The conclusion is “and inhibits NFKB” rather than via.

3. The methodology must make it clear whether NC is a vector with no insert or a vector with some other insert like GFP, or whatever.

Experimental design

Clear information on NC and OE vectors in the Methods section is essential.

Validity of the findings

Title and conclusion that effect on steatosis in mice is “via” NFKB is not supported by the data.

Reviewer 3 ·

Basic reporting

I found this manuscript is improved.

Experimental design

It is well written but needs to be arranged with numbers for the subtitles.

Validity of the findings

Good

Additional comments

Nothing to add

Reviewer 4 ·

Basic reporting

the article was well written and clear. the structures of the article was well-designed.

Experimental design

The aim of this study was clear. Detailed and vigirous experiments were conducted to illutrate the propose.

Validity of the findings

the data were robust.

---

## Round 0.3 · accepted · Accept

I believe the authors have revised the manuscript based on the comments from Reviewer 2. Considering the positive comments from the other three reviewers, I would suggest that the paper may be acceptable for publication at the current stage.